# Multidimensional conservation analysis decodes the expression of conserved long noncoding RNAs

Qiuzhong Zhou[1], Yuxi Jiang[2] , Chaoqun Cai[2], Wen Li[2], Melvin Khee-Shing Leow[1,3,4,5] , Yi Yang[6], Jin Liu[6], Dan Xu[2] , Lei Sun[1,7]

Although long noncoding RNAs (lncRNAs) experience weaker evolutionary constraints and exhibit lower sequence conservation than coding genes, they can still conserve their features in various aspects. Here, we used multiple approaches to systemically evaluate the conservation between human and mouse lncRNAs from various dimensions including sequences, promoter, global synteny, and local synteny, which led to the identification of 1,731 conserved lncRNAs with 427 high-confidence ones meeting multiple criteria. Conserved lncRNAs, compared with non-conserved ones, generally have longer gene bodies, more exons and transcripts, stronger connections with human diseases, and are more abundant and widespread across different tissues. Transcription factor (TF) profile analysis revealed a significant enrichment of TF types and numbers in the promoters of conserved lncRNAs. We further identified a set of TFs that preferentially bind to conserved lncRNAs and exert stronger regulation on conserved than non-conserved lncRNAs. Our study has reconciled some discrepant interpretations of lncRNA conservation and revealed a new set of transcriptional factors ruling the expression of conserved lncRNAs.

## Introduction

The transcriptional landscape is far more complex than initially thought, as most of the genomic sequence is pervasively transcribed into a wide variety of RNAs (Berretta & Morillon, 2009; Djebali et al, 2012). A substantial portion of RNA transcripts does not encode proteins and is regarded as noncoding RNAs. A major category of noncoding RNAs is long noncoding RNAs (lncRNAs), loosely defined as the long RNA transcripts (>200 bp) with no apparent protein-coding capacity (Derrien et al, 2012; Quinn et al, 2016). Although lncRNAs were once deemed to be the "transcriptional noise" or abandonpted RNA for a long time (Ponjavic et al, 2007), the past 15 yr have witnessed a significant rise of interest in lncRNAs as numerous lncRNAs are emerging as regulators in various biological processes (Wilusz et al, 2009; Fatica & Bozzoni, 2014; Knoll et al, 2015; Butler et al, 2019).

Because only a very small fraction of lncRNAs have been functionally characterized, the study of lncRNA conservation through comparative genomic analysis provides an important piece of evidence for their functional implication (Guttman et al, 2009; Mercer et al, 2009; Chen et al, 2016; Quinn et al, 2016; Tichon et al, 2016; Ulitsky, 2016; Sarropoulos et al, 2019; Darbellay & Necsulea, 2020). Unlike protein-coding genes, the primary sequences of lncRNAs encounter less evolutionary constraints so most lncRNAs exhibit poor sequence similarity even between closed species (Guttman et al, 2010; Ulitsky et al, 2011; Kutter et al, 2012; Ulitsky & Bartel, 2013; Necsulea et al, 2014; Hezroni et al, 2015). However, lncRNAs could be conserved at additional dimensions such as secondary structures, function, promoter sequences, and syntenic loci (i.e., genes located in a conserved order along the chromosomes) (Diederichs, 2014; Sharma & Carninci, 2020). Assessing conservation with different dimensions inevitably result in distinct sets of conserved lncRNAs (Guttman et al, 2010; Nielsen et al, 2014; Hezroni et al, 2015; Hezroni et al, 2017; Amaral et al, 2018; Ding et al, 2018; Guo et al, 2020; Ruan et al, 2020). For instance, some studies have suggested that conserved lncRNAs tend to be more abundant, contain more TE (transposable element), and exhibit lower tissue specificity (Nielsen et al, 2014; Hezroni et al, 2015), but these features were not always found in other studies (Guttman et al, 2010; Hezroni et al, 2017; Ding et al, 2018), presenting a significant challenge for understanding the properties and functions of conserved lncRNAs.

Over the past decade, studies on lncRNA have consistently shown that lncRNAs are expressed at lower abundance than protein-coding genes but with higher tissue specificity (Guttman et al, 2010; Cabili et al, 2011). The distinct features between lncRNA and mRNA expression suggest that the *trans* regulators and *cis* "codes" ruling lncRNAs transcription are likely different from those

[1]Cardiovascular & Metabolic Disorders Program, Duke-NUS Medical School, Singapore, Singapore    [2]Zhejiang Provincial Key Laboratory of Medical Genetics, Key Laboratory of Laboratory Medicine, Ministry of Education, School of Laboratory Medicine and Life Sciences, Wenzhou Medical University, Wenzhou, China    [3]Lee Kong Chian School of Medicine, Nanyang Technological University, Singapore, Singapore    [4]Yong Loo Lin School of Medicine, National University of Singapore, Singapore, Singapore    [5]Department of Endocrinology, Tan Tock Seng Hospital, Singapore, Singapore    [6]Program in Health Services & Systems Research, Duke-NUS Medical School, Singapore, Singapore    [7]Institute of Molecular and Cell Biology, Singapore, Singapore

Correspondence: danxuwzmc@qq.com; sun.lei@duke-nus.edu.sg

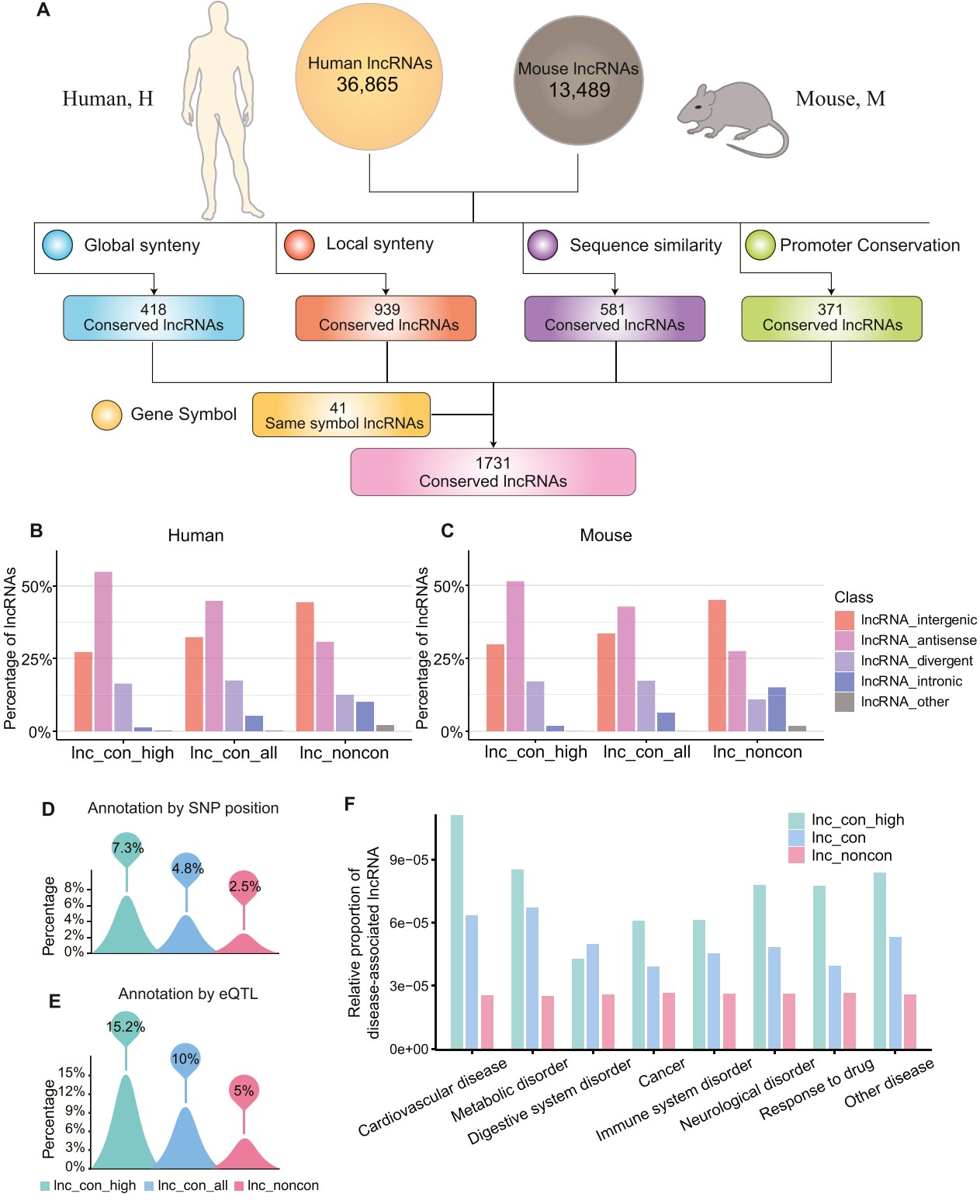

**Figure 1. Identification of conserved lncRNAs between human and mouse.**
**(A)** Conserved lncRNAs between human and mouse. We identified the conserved lncRNAs by the method of gene symbol, global synteny, local synteny, sequence similarity, and promoter conservation, separately (detailed in the Materials and Methods section). **(B, C)** The distribution of antisense, intergenic, intronic, and divergent lncRNAs in the non-conserved, conserved, and high-confidence lncRNAs for human (B) and mouse (C). **(D)** Percentage of disease-associated lncRNAs annotated by the position of SNP (single-nucleotide polymorphism) in the non-conserved, conserved, and high-confidence lncRNAs. If an lncRNA had a disease-associated SNP in its exon, this lncRNA is defined as the disease-associated lncRNAs. **(E)** Percentage of disease-associated lncRNAs annotated by eQTL (expression uantitative trait loci) in the

ruling protein-coding genes. Indeed, in silico transcription factor (TF)–binding site analysis on tetrapod-conserved lncRNAs uncovered different types of TFs with preferential binding on lncRNA versus protein-coding promoters (Necsulea et al, 2014). Furthermore, intergenic lncRNA promoters have been shown to contain less complex TF motif profiles and lower TF motif density than protein-coding promoters, suggesting that the diversity and density of TF motifs in promoters positively contribute to the transcription activity at the expense of tissue specificity (Mattioli et al, 2019). However, it remains unknown whether conserved lncRNAs and non-conserved lncRNAs bear different TF-binding profiles and thus are controlled by different sets of TFs.

In this study, we integrated multiple approaches to evaluate lncRNA conservation from different dimensions, leading to the identification of a collection of 1,731 conserved lncRNAs between human and mouse. The conserved and non-conserved lncRNAs exhibit a significant difference in their abundance, transcript number, tissue specificity, etc. Interestingly, lncRNA conservation is associated with the TF motif profiles in their promoter regions in terms of both diversity and density. In line with this observation, we identified a set of TFs that regulate the transcription of conserved lncRNAs and verified their function by gain- and loss-of-function analysis. Taken together, our results have reconciled many discrepant interpretations of lncRNA conservation and uncovered a regulatory mechanism governing the expression of conserved lncRNAs.

# Results and Discussion

### Systematic identification of conserved lncRNAs between human and mouse

To provide a comprehensive catalog of conserved lncRNAs, we used a variety of methods established in earlier reports to evaluate the global synteny, local synteny, sequence conservation, and promoter conservation of the lncRNAs in mouse and human (Fig 1A) (Necsulea et al, 2014; Hezroni et al, 2015; Amaral et al, 2018; Ding et al, 2018; Darbellay & Necsulea, 2020). First, the global synteny approach assesses lncRNA conservation based on their whole-genome sequences alignment (Hezroni et al, 2017; Darbellay & Necsulea, 2020). We lifted the genomic positions of lncRNAs from one species to the corresponding positions in the other species based on the genome synteny alignment (Fig S1A). If a mouse lncRNA and a human lncRNA superimpose on each other after lift-over, these two lncRNAs are considered to be a conserved pair. Second, the local synteny approach defines lncRNA conservation based on their nearest protein-coding genes (Ding et al, 2018; Bryzghalov et al, 2020). If an lncRNA's nearest flanking coding genes in mouse can simultaneously match the flanking genes of an lncRNA in human, this lncRNA is considered conserved in its local synteny (Fig S1B). Third, the sequence conservation approach determines lncRNA conservation using sequence

homologous alignment (Fig S1C) (Necsulea et al, 2014). The lncRNA–lncRNA pairs in mouse and human with phastCons score >0.58 were defined as sequence conserved lncRNAs. Finally, lncRNAs with highly conserved promoters between human and mouse were considered promoter-conserved lncRNA (Fig S1D) (Amaral et al, 2018).

Using the approaches described above, we identified 418, 939, 581, and 371 lncRNAs as globally, locally, sequence, and promoter-conserved lncRNAs, respectively (Figs 1A and S1A–D and Table S1). In addition, we noticed that 41 lncRNAs have the same gene symbols (Yates et al, 2017), and all of them are established as conserved lncRNAs in other reports (Russell et al, 2006; Hezroni et al, 2015; Amaral et al, 2018), thus these lncRNAs are also included in our list (Fig 1A and Table S1). Overall, we identified a total of 1,731 conserved lncRNAs between human and mouse (Table S2), Of these, ~53% of them overlapping with the conserved lncRNAs reported in the previous studies (Hezroni et al, 2015; Amaral et al, 2018; Sarropoulos et al, 2019). 427 lncRNAs pass the assessment of at least two approaches and are thereby referred to as high-confidence lncRNAs (Table S2). As expected, many previously reported conserved lncRNAs, such as *XIST*, *HOTAIR*, and *MALAT1*, were found in the high-confidence category (Table S2).

Based on the proximity of lncRNAs to nearby coding genes, lncRNAs can be broadly classified as intergenic, antisense, divergent, intronic, and other types. The intergenic and antisense types account for the most of the lncRNAs in both conserved and non-conserved lncRNAs, with antisense type comprising the highest proportion in conserved ones (Figs 1B and C and S2A and B). The percentage of antisense type is considerably higher in conserved (~45% for human and ~43% for mouse) than non-conserved (~31% for human and ~28% for mouse) lncRNAs and is even higher in the high-confidence lncRNAs (~55% for human and ~52% for mouse) (Fig 1B and C).

We speculated that the higher proportion of antisense lncRNAs in conserved group may biasely drive stronger sequence conservation for the conserved lncRNAs as a whole group due to the overlapping nature between antisense lncRNAs and coding genes (Fig S2C). To address this question, we compared the sequence conservation between antisense lncRNAs from conserved and non-conserved groups. The antisense lncRNAs from the conserved group still exhibit stronger sequence conservation than those from non-conserved group (Fig S2D). Therefore, the sequence conservation observed in the conserved group is not only merely contributed by a higher prevalence of coding gene–overlapping transcripts but also by their intrinsic sequence conservation nature.

Moreover, in human, 6.72% antisense lncRNAs are conserved whereas 3.79% "nonoverlapping" lncRNAs are conserved. In mouse, 18.75% antisense lncRNAs are conserved whereas 10.48% "nonoverlapping" lncRNAs are conserved. The antisense lncRNAs appear to be more conserved than nonoverlapping lncRNAs, likely due to their mRNA-overlapping nature. However, the vast majority of antisense lncRNAs, 93.28% in human and ~89.52% in mouse, are not

non-conserved, conserved, and high-confidence lncRNAs. If the expression of an lncRNA linked to a disease-associated eQTL, this lncRNA is defined as the disease-associated lncRNA. **(F)** The relative proportion of disease-associated lncRNA in the high-conserved, conserved, and non-conserved lncRNAs. We assigned the disease-associated lncRNAs to differential disease based on the disease category of NHGRI-EBI GWAS Catalog. The proportion was normalized by dividing the number of disease-associated conserved/non-conserved lncRNA by the total of disease-associated lncRNA and total of conserved/non-conserved lncRNAs.

able to pass the criteria of conservation assessment, indicating that mRNA-overlapping is not a predominant factor for lncRNA conservation in our assessment pipeline. Therefore, the mRNA-overlapping nature of antisense lncRNAs is unlikely to introduce significant bias into our downstream analysis.

### Conserved lncRNAs are preferentially connected to human diseases

To examine the disease association of non-conserved and conserved lncRNAs, we used the GWAS (genome-wide association study) and eQTL (expression quantitative trait loci) data to identify the disease-associated lncRNAs. If an lncRNA has a disease-associated SNP (single-nucleotide polymorphism) in its exon or is linked to a disease-associated eQTL, it is considered as a disease-associated lncRNA. We identified 963 (83 conserved and 880 non-conserved lncRNAs) and 1,914 (173 conserved and 1,741 non-conserved lncRNAs) disease-associated lncRNAs based on the SNP-position and eQTL, respectively (Tables S3–S6). In both methods, the lncRNAs with higher conservation are more likely to associate with diseases (Figs 1D and E and S3A and B), supporting a connection between lncRNAs' conservation and their functional importance. It is notable that the conserved lncRNAs are more related to cardiovascular disease and metabolic disorder than others (Fig 1F), suggesting a potential role for conserved lncRNAs in these diseases.

Insulin and glucose responses are closely related to cardiovascular and metabolic disease. To further inspect the functional importance of conserved lncRNA for these diseases, we analyzed the published datasets of insulin-stimulated adipocytes and high gluocse–treated bone marrow stem cells (Degirmenci et al, 2019; Lao et al, 2022). In both datasets, a higher proportion of conserved lncRNAs respond significantly to nutrient stimuli than non-conserved lncRNAs, suggesting a more crucial role of the conserved lncRNAs in the insulin and glucose response (Fig S3C and D).

### Characterize the gene structure difference between conserved and non-conserved lncRNAs

To systemically characterize the gene structure difference between conserved and non-conserved lncRNAs, we investigated their transcript numbers per gene, exon numbers, gene length, and transposable elements (TEs) insertion. The extent of conservation is positively associated with transcript number (Figs 2A and S4A), exon number (Figs 2B and S4B), and gene length (Figs 2C and S4C) in both species. The conserved lncRNAs contain higher transcript and exon numbers and longer gene than non-conserved lncRNAs (Fig 2A–C); the high-confidence lncRNAs, compared with the conserved lncRNAs, have a higher number of transcripts and exons and longer genes. However, even the high-confidence lncRNAs have substantially fewer transcripts and exons and shorter genes in comparison with coding genes (Fig 2A–C). To preclude the bias due to the imbalanced sample size between conserved and non-conserved lncRNAs, we conducted 1,000 random resampling analyses with a sample size of 500 and observed the similar results (Figs 2A–C and S4D), which reinforces the structural characteristic difference observed above.

Transposable elements (TEs) often replicate themselves by inserting their copies into genomes. It has been reported that TEs can make a major contribution to the origin and diversification of lncRNAs in vertebrate animals (Kapusta et al, 2013). To examine whether the presence of TEs may influence the extent of conservation, we calculated the percentage of TE-derived sequences in our lncRNA repertoire using RepeatMasker (v4.07). We found that TE-sequences are preferentially embedded in non-conserved lncRNAs and are least present in high-confidence lncRNAs (Fig 2D). This distribution pattern is consistent in both exon and intron regions (Figs 2D and S4E). Therefore, the prevalence of TEs in lncRNAs is negatively associated with lncRNAs' conservation, suggesting that TEs are significant driving factors for the rapid evolution of lncRNAs.

### Conserved lncRNAs have a higher expression correlation with their nearby genes

Because lncRNAs often act as the cis-regulatory elements in mammalian gene regulation (Yan et al, 2017), we investigated whether the conserved lncRNAs have stronger association with its nearby gene using 272 (human) and 136 (mouse) samples from a variety of sources (Poitou et al, 2015; Li et al, 2017; Varemo et al, 2017; Ramnath et al, 2018). Based on the expression correlation between lncRNAs and nearby genes within the flanking 50 kb, lncRNAs were divided into positively and negatively correlated groups. Regardless of conservation, the vast majority of lncRNAs are positively correlated with their nearby genes (Figs 2E and S5A). We further classified these nearby genes into two types, lncRNA-overlapping and lncRNA-nonoverlapping genes (Fig 2F). We found that the conserved lncRNAs, compared with non-conserved lncRNAs, exhibit stronger correlations with their overlapping genes, and the high-confidence lncRNAs have the strongest correlation (Fig 2G). Such a correlation may be due to a looser and more opening chromatin structures near the conserved lncRNA loci that facilate the access of regulatory factors to this region and lead to stronger co-regulations for the genes in proximity to the lncRNAs. It is worth noting that conserved lncRNAs exhibit a higher correlation only with their nearby gene within their overlapping genes (Fig S5B), revealing a role of the lncRNA–mRNA distance in maintaining their co-regulation and/or regulatory interactions during evolution.

### Conserved lncRNAs are more widely expressed than non-conserved lncRNAs across tissues

To investigate whether the conservation of lncRNAs is associated with their abundance and dynamic distributions across different tissues, we analyzed lncRNA expression profiles in humans and mice using RNA-seq data from multiple tissues. The detectable percentage (Fig 3A and B) and abundance of conserved lncRNAs (Fig 3C and D) are generally higher than those of non-conserved lncRNAs across most examined tissues. Moreover, the high-confidence lncRNAs exhibit even higher detectable percentage and abundance than the conserved ones (Fig 3A–D). The widespread expression nature of conserved lncRNAs is also observed in all subcategories of conserved lncRNAs (Fig S6A–D). We next sought to inspect the relationship between tissue specificity and lncRNA conservation. We used the maximum fractional expression

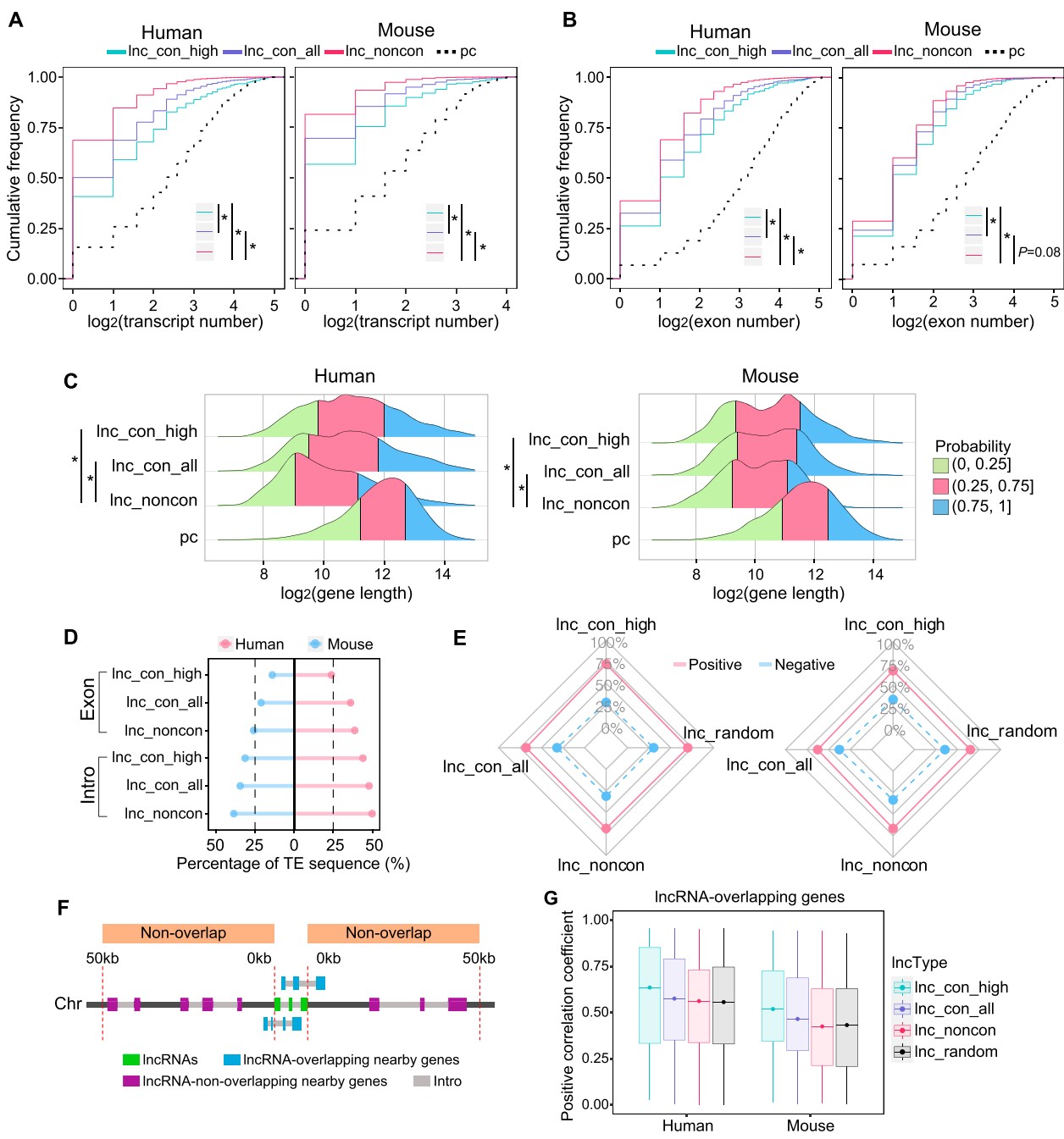

**Figure 2. Global characteristics of the non-conserved, conserved, and high-conserved lncRNAs.**
**(A, B, C)** Transcript number, (B) exon number, and (C) gene length among the non-conserved, conserved, and high-confidence lncRNAs in human and mouse.
**(D)** Percentage of TE sequence in the non-conserved, conserved, and high-confidence lncRNAs. **(E)** Percentage of positively and negatively correlated pairs between lncRNA and its nearby genes within the flanking 50 kb. Lnc_random: 1,000 lncRNA–mRNA random pairs (null model). **(F)** Schematic of overlapping and nonoverlapping nearby genes of lncRNAs. Only genes located within 50 kb up- and down-stream of lncRNAs are considered. **(G)** The correlation coefficient between the lncRNAs and its overlapping genes for the non-conserved, conserved, and high-confidence lncRNAs. We randomly resample 1,000 lncRNA–mRNA pairs in the same distance as the null model. The box covers a range from the 25th percentile (lower quartile) to the 75th percentile (upper quartile), which refers to the interquartile range (IQR). The dot in the box represents the 50th percentile (median). The upper whisker is the largest observation less than or equal to upper quartile +1.5*IQR. The lower whisker is the smallest observation greater than or equal to lower quartile −1.5*IQR. (* in A and B, *P*-value < 0.05, Kolmogorov–Smirnov test; * in G, *P*-value < 0.05, Mann–Whitney *U* test).

to define the tissue-specific score and found that both conserved and high-confidence lncRNAs exhibit lower tissue specificity than non-conserved lncRNAs in both species (Fig 3E and F). More quantitatively, regardless of the threshold used to define tissue specificity, both the conserved and high-confidence lncRNAs have a lower percentage of tissue-specific lncRNAs (Fig 3G and H). Thus, the extent of lncRNA conservation is generally associated with their widespread expression across different tissues.

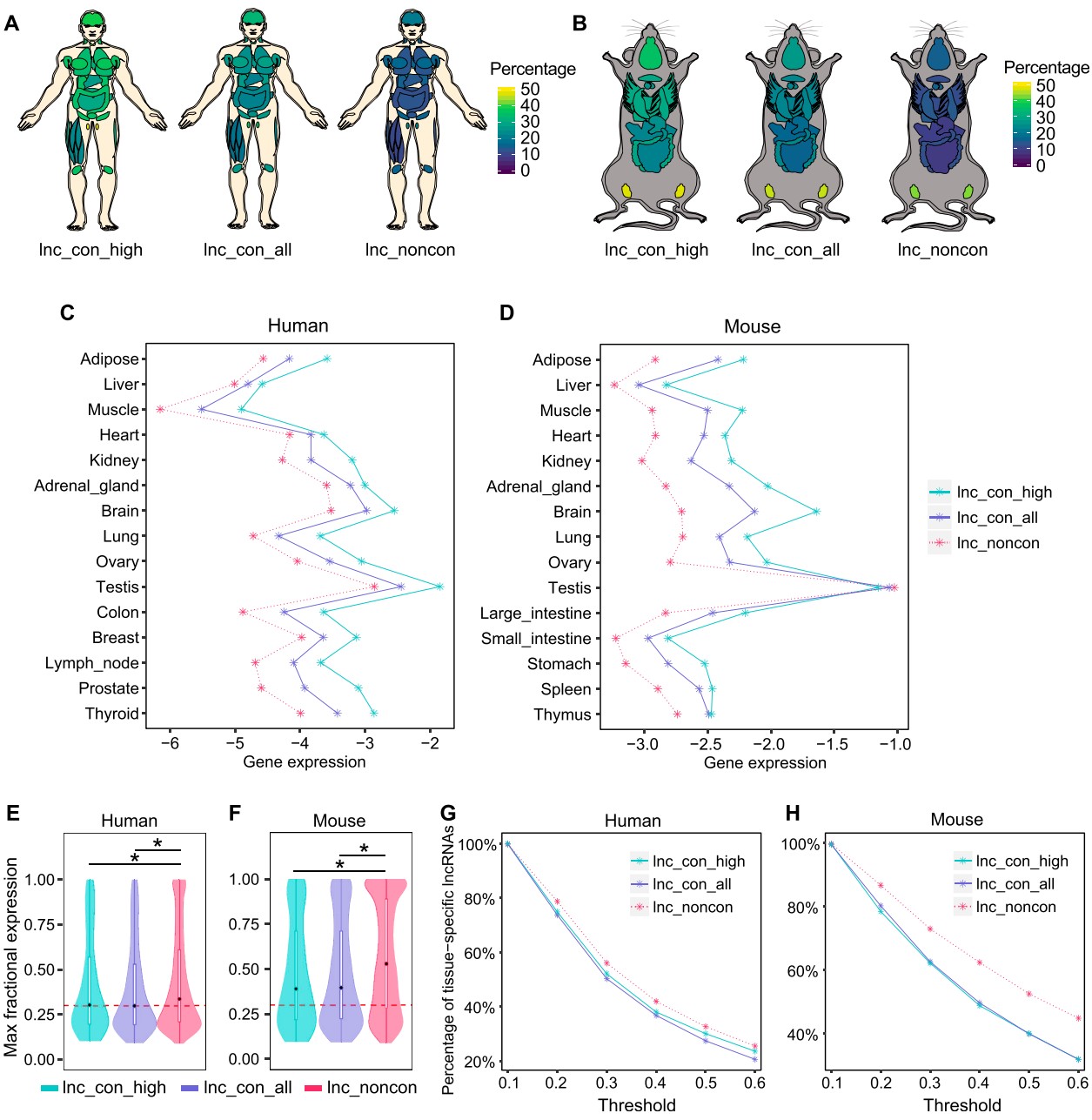

**Figure 3. Conserved lncRNAs are more widely expressed than non-conserved lncRNAs across tissues.**
**(A, B)** The expression breadth of the non-conserved, conserved, and high-confidence lncRNAs in human and mouse. The lncRNA with the expression >0.5 CPM is defined as the detectable lncRNA in each tissue. The color bar indicates the percentage of the lncRNAs detectable in each tissue. **(C, D)** Gene expression of non-conserved, conserved, and high-confidence lncRNAs in (C) human and (D) mouse across all examined tissues. **(E, F)** The max fractional expression of the non-conserved, conserved, and high-confidence lncRNAs in (E) human and (F) mouse. The embedded box represents a range of max fractional expression from the 25th percentile (lower quartile) to the 75th percentile (upper quartile). The back dot in the embedded box represents the 50th percentile (median). **(G, H)** The percentage of tissue-specific lncRNAs under the different threshold for the non-conserved, conserved, and high-confidence lncRNAs in (G) human and (H) mouse. The x-axis indicates the max fractional expression used to define the tissue-specific lncRNAs and the y-axis indicates the corresponding percentage of lncRNAs considered as tissue-specific lncRNAs given the threshold in x-axis. (*P-value < 0.05, Mann–Whitney *U* test).

## The promoters of conserved lncRNAs contain more complex transcription factor–binding motif (TFBM) profiles

Given the distinct expression patterns observed between conserved and non-conserved lncRNAs, we sought to investigate whether they are subject to distinct transcriptional regulation. Using the Find Individual Motif Occurrences (Grant et al, 2011), we analyzed the TFBMs in the promoter sequences of the conserved and non-conserved lncRNAs to calculate the TFBM number and types in each promoter. Both the TFBM types and numbers are significantly higher in the promoters of

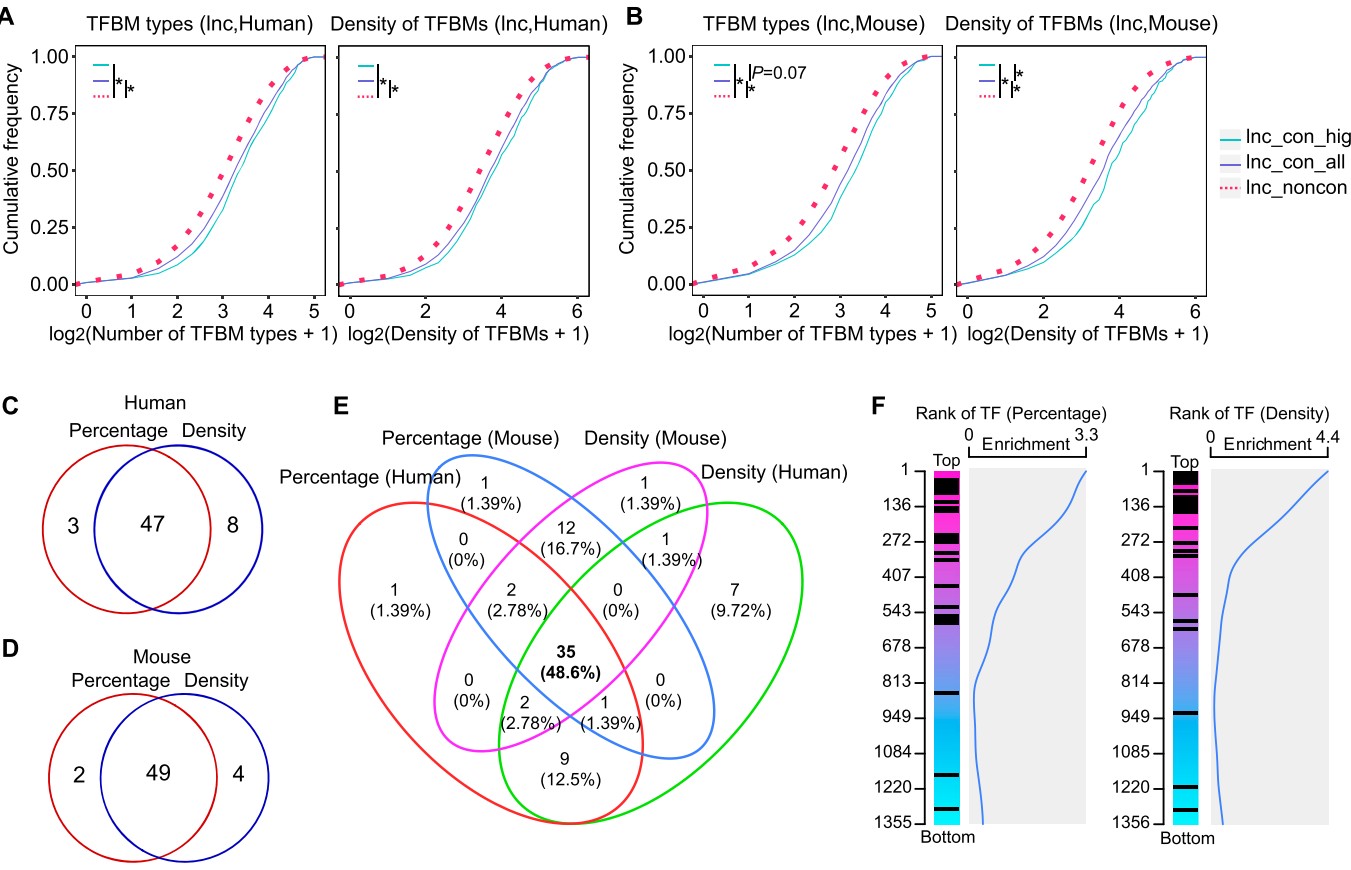

**Figure 4. Transcription factor binding motifs are enriched in the promoters of conserved lncRNAs.**
**(A, B)** The cumulative distribution of lncRNAs whose promoters contain the corresponding types of TFBMs (transcription factor–binding motif) in x-axis (left) or the corresponding numbers of TFBMs in x-axis (right) in (A) human and (B) mouse. **(C, D)** Number of TFBMs with an enrichment in the promoters of conserved lncRNAs in (C) human and (D) mouse. The red circle represents the TFBMs identified by the percentage of TFBMs (detailed in the Materials and Methods Section), whereas the blue circle represents the TFBMs identified by the density of TFMBs (detailed in the Materials and Methods Section). **(E)** The overlapping between two TFBM sets identified by two different approaches (percentage and density) in both mouse and human. **(F)** The rank of TFs based on their preferential binding to conserved lncRNAs. TF-binding sites are derived from ChIP-seq data from CistromeDB. The extent of preference of a specific TF is determined by the ratios between the percentage of conserved lncRNAs bearing binding sites of this TF and the percentage of non-conserved lncRNAs bearing binding sites of the same TF (left), and also determined by the ratios between the binding site density of this TF in conserved lncRNAs and that in non-conserved lncRNAs (right). The distribution of the motif-corresponding TFs predicted to preferentially bind to conserved lncRNAs based on motif analysis are marked as black lines in the ranking list. The density distribution of these motif-corresponding TFs is indicated by the blue line.

conserved lncRNAs, with the highest in the high-confidence lncRNAs (Fig 4A and B). To confirm our findings with experimental evidence, we used 11,213 (Human) and 9,038 (Mouse) ChIP-seq data from CistromeDB to investigate the number of TF types and TF-binding sites (TFBSs) in the promoters of lncRNAs, which reveals a consistent and more striking pattern (Fig S7A and B). Therefore, both the *cis* "code" and *trans* TF-binding analysis support that the TFBM complexity and density in lncRNAs' promoters contribute to the distinct expression patterns between conserved and non-conserved lncRNAs.

## A set of TFs preferentially bind to the promoters of conserved lncRNAs

The discovery of different TFBM complexity in the promoters of conserved versus non-conserved lncRNA encourages us to further investigate whether there exists a distinct set of TFs governing the conserved lncRNAs expression. For this purpose, we used two bioinformatic approaches to identify TFBMs preferentially embodied in the promoters of conserved lncRNAs. First, we compared the percentage of promoters bearing a specific TFBM in conserved and non-conserved lncRNAs, which led to the discovery of 50 TFBMs in human and 51 TFBMs in mouse with higher prevalence in the conserved lncRNAs (Fig 4C and D). In the second approach, we calculated the TFBM density in lncRNAs' promoters by dividing the sum of specific TFBMs by the number of lncRNAs. We identified 55 TFBMs in human and 53 TFBMs in mouse with higher TFBM density in the conserved lncRNAs' promoters (Fig 4C and D). These two approaches result in two sets of highly overlapping candidates (Fig 4C and D), attesting to the validity of our approaches. Interestingly, the TFs corresponding to these TFBMs enriched in conserved lncRNAs' promoters display a less tissue-specific pattern across tissues than non-preferential TFs (Fig S8A–D), which is consistent with the more widespread nature of conserved lncRNAs. Our findings further suggest these TFs are more likely to regulate the expression of conserved lncRNAs.

We focused on the 35 TFBMs that pass the selection criteria by both approaches in mouse and human and further analyzed their corresponding 34 TFs (Fig 4E and Table S7). To seek the experimental supports for these TFs, we examined the distribution of these 34 TFs in a ranking list where all TFs are ranked by their preferential binding to conserved lncRNAs in ChIP-seq analysis. Similar to the aforementioned TFBM analysis (Fig 4E), the preferential binding of a TF on conserved lncRNAs was assessed (1) by the percentage of lncRNAs bearing the TFBSs and (2) by the density of the TFBSs in lncRNAs' promoters (Fig 4F). Regardless of the assessment standard, the 34 motif-corresponding TFs are distributed toward the top region in the ranking list (Fig 4F), revealing a consistency between motif-based bioinformatic analysis and the ChIP-seq evidence. We referred to this set of TFs as conserved lncRNA–binding TFs (CLB_TFs) below (Table S7). Taken together, our results indicate that there indeed exists a set of TFs with more binding sites in the promoters of the conserved lncRNAs.

### CLB_TFs are more likely to modulate the expression of conserved lncRNAs

To determine whether the CLB_TFs can have a greater impact on the expression of conserved lncRNAs, we overexpressed CLB_TFs in the 293T cells and performed the RNA-seq to detect the consequential transcriptome alterations for the conserved and non-conserved lncRNAs. In parallel, we overexpressed four other TFs (HOXA11, RORA, TFCP2, and NKX2-8) that do not show any preferential binding as negative controls. We selected four CLB_TFs candidates (NRF1, E2F4, EGR1, and ZBTB7A) based on their rank in ChIP-seq data. In addition, we also conducted the null distribution test of expected versus observed observation to confirm these CLB_TFs candidates. The distribution of the proportions of lncRNAs bearing binding motif for each TF is used as the null distribution. We found that the TF with motif enrichment, such as NRF1, E2F4, EGR1, and ZBTB7A, and binding percent of conserved lncRNAs are located in the top 5% of the null distribution (Fig S9A). Consistently, these four CLB_TFs in general induced a higher number of differentially expressed genes—both lncRNAs and mRNAs—than the other four control TFs, with most of these DGEs up-regulated (Fig S9B and C), which support a stronger effect of CLB_TFs on gene expression.

To validate our bioinformatic prediction, we examined the interactions between the presence of TFBMs in lncRNA promoters and the corresponding lncRNA expression changes by TFs. We found a higher prevalence of TFBM-containing genes in the TF-regulated than non-regulated lncRNAs (Fig 5A) and a higher proportion of differentially regulated lncRNAs in TFBM-containing lncRNAs than TFBM-free lncRNAs (Fig 5B), supporting the validity of our bioinformatically predicted TFBMs. We next examined the effect of CLB_TF overexpression on conserved and non-conserved lncRNAs and found that these CLB_TFs indeed generated greater gene expression alterations for conserved lncRNAs than non-conserved lncRNAs (Fig 5C) whereas the control non-CLB_TFs did not (Fig S9D).

To assess the effect of these CLB_TFs on lncRNA expression in vivo, we analyzed the mouse RNA-seq data of *Nrf1* knockout in the retina (Kiyama et al, 2018). Consistent with our in vitro results, the loss-of-*Nrf1* results in larger gene expression changes for conserved lncRNAs compared with non-conserved lncRNAs, reinforcing

that those lncRNAs that are regulated by some TFs are more evolutionarily conserved (Fig 5D). Interestingly, despite in different cellular contexts and species, the *Nrf1* knockout–caused lncRNA expression alterations correlate negatively with those from *Nrf1* overexpression (Fig 5E), indicating that the regulation of Nrf1 on lncRNA expression can persist through specie barrier and cross different cell types. Furthermore, more binding sites of *NRF1* in the conserved lncRNA promoters can be readily detected in ChIP-seq analysis (Fig 5F and G). Taken together, our results have proven the existence of a set of TFs that can be more likely to bind and regulate the expression of conserved lncRNAs over the non-conserved ones.

## Materials and Methods

### Identification of conserved lncRNAs between human and mouse

We collected the lncRNAs annotation from GENCODE (Version 28lift37 for human and Version M17 for mouse) (Frankish et al, 2019), FANTOM5 (Hon et al, 2017), and our previous study (Alvarez-Dominguez et al, 2015; Ding et al, 2018). We filtered the lncRNAs overlapped with protein-coding mRNAs in the same strand and removed the redundancy of lncRNAs (Amaral et al, 2018). To provide a comprehensive category of conserved lncRNAs, we used multiple methods to identify the conserved lncRNAs between human and mouse (Fig S1A–D).

First, if the lncRNA bears the same gene symbol between human and mouse and is established as conserved lncRNAs in the previous study, the lncRNA was considered as the symbol-conserved lncRNAs.

Second, we used the globally syntenic regions which were generated by the whole-genome alignment between human and mouse to identify the conserved lncRNAs (Li et al, 2017). We downloaded the genome synteny alignment of mouse-to-human and human-to-mouse (chain files) created by mouse-to-human and human-to-mouse genome alignment, respectively. We lifted the mouse lncRNAs to human genome (mouse-mapped lncRNAs) and the human lncRNAs to mouse genome (human-mapped lncRNAs) based on the genome synteny alignment of mouse-to-human and human-to-mouse (chain files) (Fig S1A). If the mouse-mapped lncRNAs overlapped with the human lncRNAs or the human-mapped lncRNAs overlapped with mouse lncRNAs, these lncRNA pairs were retained. Only the lncRNA–lncRNA pairs which were commonly identified by the mouse-to-human and human-to-mouse were defined as the globally conserved lncRNAs (lnc_con_global) (Fig S1A).

Third, we used the local synteny approach to identify the conserved lncRNAs based on their nearest protein-coding genes (Amaral et al, 2018; Ding et al, 2018; Bryzghalov et al, 2020). In this method, we screened the lncRNAs whose nearest protein-coding genes were conserved in the 500 kb of upstream and downstream regions. The conserved protein-coding genes were obtained from the Ensembl database (Fig S1B).

(a) If only one lncRNA was present between the upstream and downstream nearest conserved protein-coding genes in human

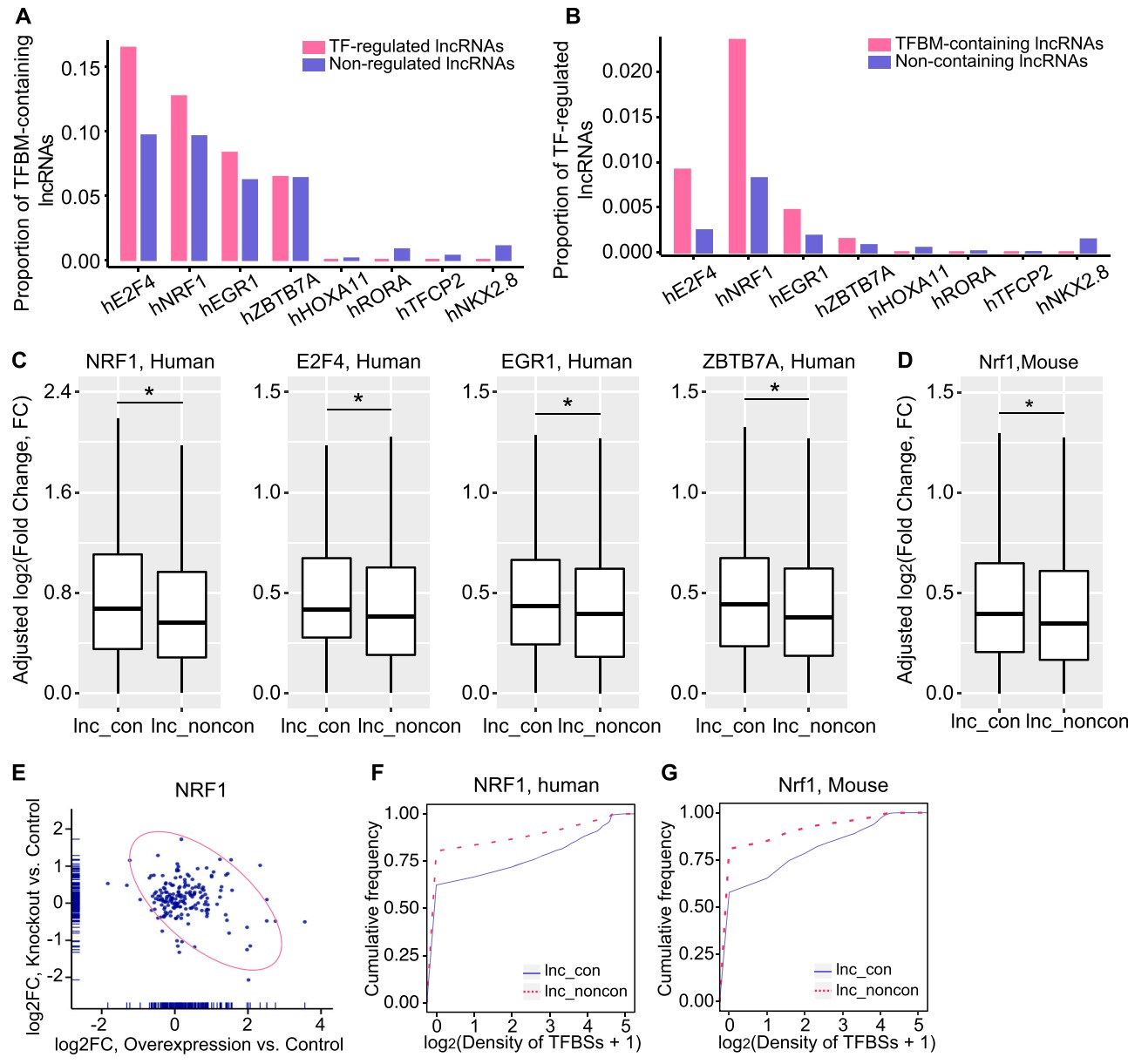

**Figure 5. Conserved lncRNAs are likely to be regulated by CLB_TFs.**
**(A)** The relative proportion of TFBM-containing lncRNAs in TF-regulated lncRNAs and the non-regulated lncRNAs. **(B)** The relative proportion of TF-regulated lncRNAs in the TFBM-containing and non-containing lncRNAs. We used a cut-off of FDR < 0.05 to identify the TF-regualted lncRNAs in the overexpression RNA-seq data. **(C)** Expression change of the conserved and non-conserved lncRNAs upon the overexpression of CLB-TFs (*NRF1*, *E2F4*, *EGR1*, and *ZBTB7A*) in 293T cells. Expression change was adjusted by the gene abundance based on a linear regression model. The box indicates the interquartile range (IQR) from the 25th percentile (lower quartile) to the 75th percentile (upper quartile). The straight line in the box represents the median value. The upper whisker is the largest observation less than or equal to upper quartile +1.5*IQR. The lower whisker is the smallest observation greater than or equal to lower quartile −1.5*IQR. **(D)** The expression change of the conserved and non-conserved lncRNAs upon *NRF1* knockout in the mouse retina. **(E)** The relationship of $\log_2$ (fold change of expression) between the overexpression versus control and knockout versus control experiment for NRF1. **(F, G)** The density of NRF1-binding sites between the non-conserved and conserved lncRNAs in (F) human and (G) mouse. The binding sites were assessed by the ChIP-seq data of NRF1. (* in A and B, *P*-value < 0.05, Kolmogorov–Smirnov test, * in G and H, *P*-value < 0.05, Mann–Whitney *U* test).

and mouse, this lncRNA pair was defined as locally conserved lncRNAs (lnc_con_local) (Fig S1B).
(b) If more than one lncRNA was present between the upstream and downstream nearest conserved protein-coding genes in at least one species, we performed the sequence alignment for these lncRNAs. The best hit for each other was also considered as the locally conserved lncRNAs (lnc_con_local) (Fig S1B).

Fourth, to discover the conserved lncRNAs with sequence conservation, we conducted the sequence homologous alignment of lncRNAs using BLAST with the E-value < 1 × 10⁻⁵ and identity ≥80% and BLAST with identity ≥80% for mouse-to-human and human-to-mouse, respectively (Fig S1C) (Necsulea et al, 2014). Only the best hit for each other was retained. In each lncRNAs pair, we used phastCons and phyloP to calculate the

conserved score between human and mouse (Siepel et al, 2005). We performed the sliding window analysis for the conserved score with a window size of 200 bp and step length of 50 bp. Only the lncRNAs pair with the conserved score > 0.58 in at least one window was defined as the sequence-conserved lncRNAs (lnc_con_seq).

Fifth, we conducted the sequence alignment for the promoters of lncRNAs from mouse-to-human and from human-to-mouse using BLAST (v2.2.28+) with parameters of max_target_seqs 1, word_size 6, evalue 1E-10, strand plus, and perc_identity 80, separately (Fig S1D) (Camacho et al, 2009; Amaral et al, 2018). The lncRNA pairs with the best fit were defined as the promoter-conserved lncRNAs (lnc_con_promoter).

Ultimately, after removing the redundancy, we identified 1,731 conserved lncRNAs between human and mouse. If the lncRNAs pass the assessment of at least two methods, they were defined as high-confidence lncRNAs between human and mouse.

### Distribution of lncRNA types in conserved and non-conserved lncRNAs

To understand the contribution of different types of lncRNAs to conserved lncRNAs, we used the tool, FEELnc (FlExible Extraction of LncRNAs) (Wucher et al, 2017) to class the lncRNAs into four types: antisense, intergenic, intronic, and divergent lncRNAs based on their position relationship with protein-coding mRNAs. The lncRNAs which are not assigned to the above types would be regarded as the other type lncRNAs. We then assessed the composition of these lncRNA types in high-confidence, conserved, and non-conserved lncRNAs, and in different subcategories of lncRNAs including globally conserved lncRNAs, locally conserved lncRNAs, sequence-conserved lncRNAs, and promoter-conserved lncRNAs.

### Evolutionary conservation analysis between the conserved and non-conserved antisense lncRNAs

We downloaded the 46-way and 60-way phastCons score from the UCSC ([https://genome.ucsc.edu/](https://genome.ucsc.edu/)) for human and mouse, respectively. We defined the average of single-nucleotide phastCons score as the conservation score for each lncRNA and compared the conservation score between the conserved and non-conserved antisense lncRNAs.

### Disease annotation of lncRNAs

We used two approaches to identify disease-associated lncRNAs. First, if an lncRNA has a disease-associated SNP (single-nucleotide polymorphism) in its exon, it will be regarded as the disease-associated lncRNAs. The disease-associated SNPs are downloaded from GWAS Catalog (Ardlie et al, 2015; Buniello et al, 2019). Second, if the expression of an lncRNA is linked to a disease-associated eQTL (expression quantitative trait loci) then this lncRNA will also be defined as the disease-associated lncRNA. The significantly associated eQTLs were obtained from the GTEx (Version 7) (Ardlie et al, 2015).

### Characterize the gene structure between the conserved and non-conserved lncRNAs

We evaluated the gene feature between the conserved and non-conserved lncRNAs including the transcript number, exon number, and gene length based on the gene annotation of human and mouse. We used RepeatMasker (v4.07) to calculate the percentage of sequence-based TE (transposable elements) content in the conserved and non-conserved lncRNAs (Kapusta et al, 2013; Hezroni et al, 2017). To confirm whether the imbalanced number between conserved and non-conserved lncRNAs biases the results during their comparisons, we conducted these comparisons by 1,000-times random resampling (sample size n = 500) and then compared the characteristic such as gene length and transcript number between the conserved and non-conserved lncRNAs.

### Expression abundance across the examined tissues

To compare the expression breadth between the conserved and non-conserved lncRNAs, we calculate the expression of lncRNAs using the Human Body Map 2 data, Mouse Body Map (Li et al, 2017), and RNA-seq of mouse adipose tissues generated from our laboratory. We mapped the human and mouse RNA-seq data to the human genome (GRCh37) and mouse genome (GRCm38) using STAR (v.2.6.0c) (Dobin et al, 2013), respectively. featureCounts (v.1.6.3) (Liao et al, 2014) was used to calculate the read counts of conserved and non-conserved lncRNAs. Only uniquely mapped reads were retained. The read counts were normalized to CPM (counts per million) using the R package, edgeR (v.3.20.9) (Robinson et al, 2010). An lncRNA with the expression >0.5 CPM was considered as the detectable lncRNAs in each tissue.

### Correlation between the lncRNA and its nearby mRNAs

We calculated the Spearman correlation coefficients between lncRNA and its nearby gene within the flanking 50 kb using 272 and 136 randomized RNA-seq samples for human and mouse, respectively. The human samples were obtained from Human Body Map 2 data (Cabili et al, 2011), adipose tissues of an obese patient cohort (Poitou et al, 2015), liver with different fibrosis stages (Ramnath et al, 2018), and primarily differentiated myotubes (Varemo et al, 2017). The mouse samples included the Mouse Body Map data (male) (Li et al, 2017) adipose, liver, and muscle with different metabolic statuses of our laboratory. In the lncRNA–mRNA correlation analysis, we used 1,000 lncRNA–mRNA random pairs within the same distance as an independent null model (Hurst et al, 2004; Ebisuya et al, 2008; Cabili et al, 2011).

### Tissue specificity of the non-conserved, conserved, and high-confidence lncRNAs

We inspect the tissue-specific score of the non-conserved and conserved lncRNAs using the gene expression from a variety of tissues based on the approach on our early studies ([Alvarez-

Dominguez et al, 2015; Ding et al, 2018). We defined the fractional expression for each lncRNA in a given tissue as the proportion of its expression against the cumulative expressions of this lncRNA across all examined tissues. We used the max fractional expression to define the tissue-specific score. To compare the percentage of tissue-specific lncRNAs between non-conserved and conserved lncRNAs, we used different thresholds (0.1, 0.2, 0.3, 0.4, 0.5, and 0.6) to define tissue-specific lncRNAs (Ding et al, 2018).

### Identification of transcription factor–binding sites in the promoter of lncRNAs

We obtained the TFBMs from JASPAR 2018 (Khan et al, 2018) and used Find Individual Motif Occurrences (Grant et al, 2011) to discover the TFBSs in the promoter (upstream: 500 bp) for the conserved and non-conserved lncRNAs. We then calculated the number of TFBM types and TFBMs in the promoter of conserved and non-conserved lncRNAs. The significance test was performed by Kolmogorov–Smirnov test ($P$-value < 0.05).

### Analysis of ChIP-seq data

We downloaded 11,348 (for human) and 9,061 (for mouse) ChIP-seq data from Cistrome database (Zheng et al, 2019). We removed the peak files with the zero file size. A total of 11,213 (for human) and 9,038 (for mouse) ChIP-seq experiments were retained for the next analysis. We calculated the number of TF types and TFBSs in non-conserved and conserved lncRNAs by overlapping the peaks of differential TFs to the position of its promoter (upstream: 500 bp). We used a Kolmogorov–Smirnov test to test the significance in the number of TF types and TFBSs between non-conserved and conserved lncRNAs.

### Identification of TFs with preferential binding to the conserved lncRNAs

To find the TFs that are more likely to regulate the conserved lncRNAs, we used two bioinformatic approaches to identify the TFBM with enrichment in the promoter of conserved lncRNAs.

1)

$$\text{PR (Percentage Rate)} = \left| \left( \frac{A}{B} + R \right) \middle/ \left( \frac{C}{D} + R \right) \right| \text{ or}$$

$$\text{PM (Percentage Minus)} = \left| \left( \frac{A}{B} \right) - \left( \frac{C}{D} \right) \right|.$$

For a specific TFBM, where A is the number of conserved lncRNAs with this TFBM, B is the total number of conserved lncRNAs, C is the number of non-conserved lncRNAs with this TFBM, and D is the total number of non-conserved lncRNAs. $\frac{A}{B}$ is the percentage of conserved lncRNAs bearing this TFBM and $\frac{C}{D}$ is the percentage of non-conserved lncRNAs with this TFBM. $R$ is the median of all TFBM's percentage including the $\frac{A}{B}$ and $\frac{C}{D}$.

If this TFBM is with either the $PR$ more than the value of 90th percentiles or $PM$ more than the value of 90th percentiles and $\frac{A}{B}$ and

$\frac{C}{D}$ are significant (FDR < 0.05, two-proportions z-test), it will be defined as the TFBM with a preference to enrich in conserved lncRNAs.

2)

$$\text{DR (Density Rate)} = \left| \frac{M + R}{N + R} \right| \text{ or DM (Density Minus)} = |M - N|.$$

For a specific TFBM, where $M$ represents the density (average) of TFBMs in the promoter of conserved lncRNAs, $N$ is the density of TFBMs in the promoter of non-conserved lncRNAs. $R$ is the median of all TFBM's density including M and N. If this TF is with either the $DR$ more than the value of 90th percentiles or $DM$ more than the value of 90th percentiles and $M$ and $N$ are statistical (FDR < 0.05, Mann–Whitney $U$ test). It will be considered as the TFBM with a preference to enrich in conserved lncRNAs. The $P$-values in (1) and (2) were adjusted by the Benjamini–Hochberg method, separately (Benjamini & Hochberg, 1995).

### Overexpression of transcription factor in the 293T cells

The ORF regions of NRF1 (NM_005011.5), E2F4 (NM_001950.4), EGR1 (NM_001964.3), ZBTB7A (NM_015898.4), HOXA11 (NM_005523.5), RORA (NM_134260.2), TFCP2 (NM_005653.4), and NKX2-8 (NM_014360.3) were cloned into lentiviral vector pLV[Exp]-EGFP-EF1A>{MSC} from VectorBuilder. These plasmids and an empty control plasmid were transfected into 293T cells (ATCC) using EndoFectin Max transfection reagent (Genecopoeia) according to the manufacturer's instruction. 48 h post-transfection, total RNAs were extracted using miRNeasy kit (QIAGEN) for downstream RNA-seq analysis.

### RNA-seq analysis of overexpression and knockout data

Total RNAs were sent to Novogene for RNA-seq library preparation and sequencing. ~40 million reads per sample were generated. We used the FastQC (v.0.11.2) (Andrews, 2010) to control the quality of the RNA-seq data and mapped the pair-end reads to the human genome (GRCh37) by STAR (v.2.6.0c) (Dobin et al, 2013). We then used featureCount (v.1.6.3) (Liao et al, 2014) to compute the read counts based on the gene annotation. Only the uniquely mapped reads were retained. CPM (counts per million) is used to normalize the raw read counts (Robinson et al, 2010). We used the R package, edgeR (v.3.30.3) to perform the RNA-seq analysis (Robinson et al, 2010). The gene with FDR < 0.05 was defined as the differentially expressed genes (TF-regulated genes). The $\log_2$ fold change of (overexpressed CPM/control CPM) was used to assess the expression change by TF. The $\log_2 FC$ (overexpressed/control) is adjusted by the expression based on the linear regression model. We downloaded the RNA-seq samples of knockout NRF1 in the retina for the mouse from Gene Expression Omnibus (GEO: GSE101550) (Kiyama et al, 2018). We used the same pipeline to perform RNA-seq analysis in mouse. To check whether the overexpression and knockout of NRF1 reveals the negative correlation, we examined the relationship of expression change between overexpression versus control and knockout versus control for the conserved lncRNAs with CPM > 0.5 in both sides.

## Data Availability

The RNA-seq raw data are available on GEO (Gene Expression Omnibus) under the accession number: GSE156270.

## Supplementary Information

## Acknowledgements

This work was supported by Singapore National Medical Research Council's Open Fund—Open Fund-Individual Research (OF-IRG) Grant: MOH-000954 (L Sun), NMRC/OFIRG/0062/2017 (L Sun) and Ministry of Education (MOE) Tier2 grant: MOE2017-T2-2-015 (L Sun), MOE2019-T2-1-025 (L Sun). This work was also supported by National Medical Research Council Singapore: MOH-000657 (Q Zhou) and National Natural Science Foundation of China (82170885). We would like to thank all the members in the laboratory of Prof. Sun for their kind assistance in this study. The computational work for this article was partially performed on resources of the National Supercomputing Centre, Singapore (https://www.nscc.sg).

### Author Contributions

Q Zhou: formal analysis and writing—original draft.
Y Jiang, C Cai, and W Li: validation.
MK-S Leow: writing—review and editing.
Y Yang and J Liu: formal analysis.
D Xu: validation and project administration.
L Sun: resources and writing—review and editing.

### Conflict of Interest Statement

The authors declare that they have no conflict of interest.

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
