## [Reviewer comments · Life Science Alliance]

Life Science Alliance

Multi-dimensional conservation analysis decodes the expression of conserved long noncoding RNAs

Qiuzhong Zhou, Yuxi Jiang, Chaoqun Cai, Wen Li, Melvin Leow, Yi Yang, Jin Liu, Dan Xu, and Lei Sun
DOI: <https://doi.org/10.26508/lsa.202302002>

Corresponding author(s): Lei Sun, Duke NUS Graduate Medical School and Dan Xu, Wenzhou Medical University

Review Timeline:

Submission Date:	2023-02-20
Editorial Decision:	2023-02-22
Revision Received:	2023-03-13
Editorial Decision:	2023-03-14
Revision Received:	2023-03-21
Accepted:	2023-03-21

Transaction Report:

Please note that the manuscript was previously reviewed at another journal and the reports were taken into account in the decision-making process at Life Science Alliance.

Referee #1 Review

Report for Author:

In this report, Zhou et al detailed a study of conserved lncRNAs between human and mouse. They first used multiple approaches to identify a list of conserved lncRNAs and subsequently compared them with the non-conserved ones to define their unique features. Specifically, they found that conserved lncRNAs have longer gene bodies, more exons and transcripts and exhibit higher levels of expression. Finally, they identified a set of transcription factors that preferentially bind to promoters of conserved lncRNAs, a finding that they experimentally validated by overexpression of selected TFs. Overall, this work represents a novel and robust analysis of lncRNA conservation, and some of the findings, such as different TFBM complexity in the promoters of conserved versus non-conserved lncRNA, is very intriguing. The mode of conservation is a fundamental question in lncRNA research, and this work provides a unique perspective. The authors' focus on human and mouse is also valuable as mouse is the most widely used model in pre-clinical studies. That being said, this work can be significantly strengthened by additional validation and clarification.

1. The functional importance of the conserved lncRNAs identified by the authors needs to be further evaluated. Of course, most lncRNAs have not been experimentally characterized and this question might be beyond the scope of this work. Still, I recommend that the authors use certain proxies of functionality to further evaluate the functional potential of these conserved lncRNAs. For example, they can search and analyze published datasets to examine if conserved lncRNAs are subject to strong regulation by cellular or physiological stimuli, which can sometimes be used an indicator of functional relevance.
2. "Thus, conserved lncRNAs exert greater influence on their nearby gene expression." This is an overstatement. Many lncRNAs may not affect the expression of their nearby genes but rather are co-regulated with them by common regulatory elements.
3. It would be interesting to examine the gene expression and tissue specificity of conserved and non-conserved protein-coding genes.
4. Figure 4G and 4I, how were protein-coding genes regulated in these analyses?

Referee #2 Review

Report for Author:

The manuscript entitled "Multi-dimensional conservation analysis reveals transcriptional codes ruling the expression of conserved long noncoding RNAs" is a multifaceted approach towards the ever emerging body of literature towards identifying conserved lncRNAs. To this end the authors employ standard synteny and expand this to include super-imposition of the loci, flanking synteny and phastcons conservation. Notably, local synteny was the most informative of the approaches. The authors find that antisense lncRNAs are most represented in the conserved lncRNAs. Next the authors compare numerous properties

associated with conserved vs non-conserved lncRNA finding that length and number of exons and transcripts are enriched in conserved lncRNAs. The authors further explore, GWAS, expression and promoter properties of conserved versus non-conserved lncRNAs. Overall, this study relies on the classifier used and potential biases there in (e.g., antisense lncRNAs representing a majority of conserved lncRNAs -- perhaps based on being located within a mRNA locus -- which by default are highly conserved relative to lncRNAs). I have some concerns on the limited number of lncRNAs that meet the authors criterion compared to ~14,000 annotated comparators. Moreover, most of the results (excluding exon and length comparisons) have very small effect sizes.

1) How much of the antisense is accounted for protein coding conservation?

1a) It seems as if the approach used in this study could be biased to protein coding loci as they would score higher in all 4 categories used to define the conserved lncRNAs. Can the authors compare the sequence conservation of the non-conserved antisense relative to the conserved antisense? Are there two clear distributions. If so, how many of the conserved antisense "exons" overlap with an mRNA exon that could drive the conservation on the sequence level relative to non-conserved antisense lncRNAs.

1b) Perhaps more convincingly, promoters are one of the most conserved features of lncRNAs and similar to mRNA promoter conservation. Do the antisense lncRNAs share the same promoters across species? Do the promoters lift over across species? What is the percent conservation of antisense promoters relative to other lncRNA classifications. The promoter sequence conservation and syntenic position by lift over would be property that would be independent of the overlapping protein coding genes conservation properties. Thus, it is suggested the further require antisense lncRNAs to have conserved promoters by sequence and synteny -- at least to the level of non-overlapping lncRNAs. How many of the non-conserved antisense also have syntenic or conserved promoters? This would indicate that the conservation metrics used by the authors miss "conserved" lncRNAs of any type if they have a conserved promoter.

1c) The number of "conserved lncRNAs" is far fewer than the annotated lncRNA loci ~15,000. Thus many of these comparisons will be biased by the ratio of conserved to non-conserved, especially considering a majority overlap mRNAs.

1d) What is the relative proportion of antisense lncRNA that are conserved versus non-conserved and how does this compare to the ratio of "non-overlapping" lncRNAs that are conserved versus non-conserved. It is very unclear the null numbers of annotations being used to determine and or biasing the analysis of "conserved lncRNAs".

2) Neighboring gene regulation:

When comparing neighboring gene-expression it is imperative that a null model is derived from expression data as the genome is known to have positional bias of local expression. Or in otherwords most gene-gene pairs show more correlation in expression based on proximity. This has been noted by several studies including (Cohen et al. 2000; Hurst et al. 2004, (Ebisuya et al. 2008)

Thus, this analysis requires a null "expected" distributions that represents random correlations of gene-pairs at the same distance being measured by the authors. Then comparing the observed correlation relative to expected. Other studies that performed the correct null distribution and testing found that mRNA pairs are equally correlated by chance as lncRNA and mRNA correlations (Cabili et al. 2011). Especially considering several antisense lncRNAs have been shown to negatively regulate the overlapping mRNA. A null distribution of correlations of random antisense pairs should also be calculated independently to control for known genomic positioning effects.

3) Transcription factor regulation of conserved lncRNAs. The effect size in this analysis is very small. Suggesting nearly as many non-conserved lncRNAs have the same motifs as those that are conserved. How many other TF motifs are enriched in non-conserved over conserved? Are these motifs enriched in conserved lncRNAs over mRNA promoters? If not it explains that these small effect sizes are essentially random compared to the number of TF motifs tested. A null distribution test of expected versus observed would be more informative to determine if these motifs truly "classify" conserved lncRNA promoters.

Minor the figure legend is incomplete : "Figure 4. Conserved lncRNAs are preferentially regulated by the transcription factor"

4) 4) Finally, it seems prudent for the authors to compare how their catalog of "conserved" lncRNAs compares to the many other studies prior (e.g., Amaral et al., Herzoni et al. and others). Below are some further suggestions for the authors. How many of these annotations overlap? What are the advantages and how can they be assessed for this approach over the others?

Report for Author:

This manuscript presents an evolutionary analysis of lncRNAs in human and mouse, coupled with *in silico* and *in vitro* analyses of transcription factor binding sites in lncRNA promoters. The authors analyze various aspects of lncRNA evolution - e.g. sequence conservation, synteny conservation, promoter conservation. They define sets of ultra-conserved, conserved and non-conserved lncRNAs and they assess the presence of transcription factor binding motifs in the promoters of these lncRNA classes. They find several transcription factors that are over-represented in conserved lncRNA promoters. For several TF candidates, they confirm with *in vitro* that over-expression of these TFs leads to differential expression for conserved lncRNAs more than for non-conserved lncRNAs.

The manuscript thus consists of two different topics: first, identification of conserved lncRNAs; second, TF binding analysis for conserved lncRNAs. The analyses presented in the "evolutionary conservation" part of the manuscript do not bring anything new - evolutionary conservation of lncRNAs has been extensively studied in the past few years (see also recent paper by Sarropoulos et al., 2019). The results presented here are in agreement with these other publications, but unfortunately it is difficult to bring anything new in this well-studied field.

The second part, regarding TF binding, is somewhat more novel, although differential TF binding patterns between conserved and non-conserved lncRNAs were previously reported, as the authors acknowledge in the text (e.g. Necsulea et al., 2014). The novelty here is that there is an *in vitro* confirmation of this observation, for several transcription factors. However, this analysis is not sufficiently developed. The only result shown is a distribution of the fold expression changes after over-expression (or knockout in mouse) of several transcription factors. Are lncRNAs significantly differentially expressed? Are the differentially expressed lncRNAs those that have the corresponding transcription factor binding motifs in their promoters? What is the expression pattern of the lncRNAs - are they all up-regulated or down-regulated? How do they compare with protein-coding genes? Are they more often antisense or transcribed from bidirectional promoters shared with protein-coding genes? These are just a few examples of analyses that could help strengthen this part of the paper.

Other comments:

- the authors often over-interpret some results. For example, they say in page 8 "... indicating that the lncRNAs overlapping with coding genes are more constrained during evolution than the lncRNAs in the intergenic regions.". There is an obvious confounding factor, which is the conservation of the overlapping coding genes. Here, sequence conservation analyses should be performed, on non-overlapping regions only - that is, on lncRNA exon regions that do not overlap with protein-coding exons or intronic splicing regulatory motifs. The co-expression between neighboring protein-coding genes and lncRNAs is likewise over-interpreted: page 11, "Thus, conserved lncRNAs exert greater influence on their nearby gene expression." Correlation does not imply causation, and for antisense overlapping lncRNAs there are other scenarios that could explain this observation (e.g., opening of the chromatin that facilitates transcription of lncRNAs). Better correlations for conserved lncRNAs than for nonconserved lncRNAs could also be explained by their broader expression patterns.

- the "global" synteny analysis (lncRNAs found next to orthologous protein-coding genes, within a 500 kb region) is too permissive. Within 500kb, there is largely enough space for numerous lncRNAs, with different evolutionary origins. This is not sufficient to claim conservation between human and mouse, without further confirmation from other analyses.

- the circular barplots in figure 1B,C are hard to read and do not bring anything to the paper. Please replace these with simple, easy to read barplots.

- figure 5G, I recommend using volcano plots instead, to show log₂ fold change (not absolute values - it is important to show up- and down-regulation) and significance levels at the same time.

- the overexpression and knockout NRF1 data are not sufficiently exploited. One would expect opposite regulation patterns for genuine NRF1 targets in this context. I would recommend showing the log₂fold change in one experiment against the other, for those genes that have 1-to-1 orthologues in human and mouse.

- I recommend removing the "symbol" conservation, this is not an objective criterion. One could have similar symbols or acronyms by chance, at least in some cases.

Referee #1 Review

Report for Author:

The authors have fully addressed all my concerns.

Referee #2 Review

Report for Author:

I appreciate the authors going above and beyond to address my concerns. I agree with their responses and the additional data to clarify some of the numbers in the study.

Referee #3 Review

Report for Author:

In my opinion, the revised manuscript submitted by Zhou and co-authors does not show enough improvement compared to the original submitted version. As I mentioned in my first review, a large part of this manuscript is dedicated to an analysis of lncRNA conservation between human and mouse, and of basic lncRNA characteristics, which does not bring anything new compared to the vast existing literature. Three out of the five main figures are dedicated to these analyses. This part of the manuscript, which is confirmatory of the existing knowledge, could easily be moved to the supplementary material. The second part of the manuscript, which examines the patterns of regulation of lncRNAs by transcription factors, has the potential to bring new insights, but needs to be consolidated and corrected in many ways before it can be considered for publication.

Unfortunately I have to repeat some of my previous remarks, which have not been answered appropriately in this revised version of the manuscript.

- The text is filled with over-statements and with confusions between correlation and causality. To cite just a few examples: "The conservation of lncRNAs influences the expression of their nearby genes" (section title; there is no possible mechanism by which the evolutionary conservation of a locus can influence gene expression); "the influence of lncRNA conservation on their nearby gene expression" (same subsection); "lncRNA conservation may affect how lncRNA react to cellular response to environmental clues"; "Our findings strongly suggest a causal linkage between the TFs' expression and their targeted lncRNAs' expression." (no, the "targeted lncRNAs" are just putative targets and the link between the expression breadth of TFs and the expression breadth of these lncRNAs could be affected by a selection bias - more details below); "The CLB_TFs preferentially impact the expression of conserved lncRNAs", or "the loss-of-Nrf1 results in larger gene expression changes for conserved lncRNAs than non-conserved lncRNAs, reinforcing its regulatory preference for conserved lncRNA " -> TFs cannot distinguish or "prefer" conserved lncRNAs over non-conserved lncRNAs; a better way to present this is that those lncRNAs that are regulated by some TFs are more evolutionarily conserved.

- The analyses often lack appropriate controls. For example, the authors confirm that conserved lncRNAs are more broadly expressed than non-conserved lncRNAs. That likely means that conserved lncRNAs also have on average higher expression levels than non-conserved lncRNAs in the cell types in which they analyze the effect of TF knockdown or overexpression. Given that the power to detect differential expression between conditions strongly depends on gene expression level, this could easily affect the differences between conserved and non-conserved lncRNAs in terms differential expression after TF knockout or overexpression. It is also more likely that the promoters of conserved lncRNAs are better defined, because full-length gene models are easier to reconstruct for highly expressed genes than for weakly expressed genes. This means that there will be larger frequencies of TF binding sites in these promoters. Overall, gene expression level differences need to be accounted for in all analyses. Another example is the authors' interpretation of the frequency of disease SNPs associated with neurological disorders - both conserved and non-conserved lncRNAs are associated with such SNPs, which the authors interpret as "suggesting an involvement of lncRNAs in neuronal development and function.". But what is the overall frequency of SNPs involved in neurological disorders compared to other diseases? It may simply be the case that these diseases have been better studied so far than other diseases.

- Most of the statements in the results are vague and no numerical details are given. For example, at the very end of the results the author say that there is a negative correlation between expression alterations observed after knockout and overexpression of NRF1. However the correlation value is not provided, nor is a p-value given for a correlation test. The corresponding scatterplot in figure 5E does not show a negative correlation, despite the ellipse that was added to the plot (the authors give no explanation on how this ellipse was defined).

- The graphical representations of the data are not often useful. The authors show circular barplots in figure 1F,G, which are impossible to read and simply make no sense - they are not pie-charts, given that the total does not sum to 100%; why use a

circular representation here? Why are the axes mirrored and the human and mouse points connected in figure 2D? Why would show a 2D representation in figure 2A instead of a simple, easy to read barplot? The boxplots do not show confidence intervals of the median. In figure 5C, the authors persist in showing the absolute log₂ fold change rather than the actual positive and negative values, which makes the graphics impossible to interpret.

- My previous comments regarding the need to examine non-overlapping exonic regions for sequence conservation analyses was disregarded, although antisense overlap with other genes can greatly affect the sequence conservation estimates. Likewise, my comment regarding the window size chosen for synteny analyses was disregarded. I recommend redoing the the synteny conservation analyses with smaller window sizes and compare the results. Alternatively, one could count how many different lncRNA loci can be found in the 500kb window chosen by the authors.

- The text needs to be carefully proofread or even rewritten. There are many grammatical and typographical errors, with some phrases impossible to understand.

February 22, 2023

Re: Life Science Alliance manuscript #LSA-2023-02002-T

Dr. Lei Sun
Duke-NUS Medical School
CVMD
8 College Road
Singapore 169857

Dear Dr. Sun,

Thank you for submitting your manuscript entitled "Multi-dimensional conservation analysis reveals transcriptional codes ruling the expression of conserved long noncoding RNAs" to Life Science Alliance. We invite you to submit a revised manuscript addressing the following Reviewer comments:

- Address Reviewer 3's 1st, 2nd and 6th detailed remark.

Thank you for this interesting contribution to Life Science Alliance. We are looking forward to receiving your revised manuscript.

Sincerely,

B. MANUSCRIPT ORGANIZATION AND FORMATTING:

Reviewer 3's 1st, 2nd and 6th detailed remark

- The text is filled with over-statements and with confusions between correlation and causality. To cite just a few examples: "The conservation of lncRNAs influences the expression of their nearby genes" (section title; there is no possible mechanism by which the evolutionary conservation of a locus can influence gene expression); "the influence of lncRNA conservation on their nearby gene expression" (same subsection); "lncRNA conservation may affect how lncRNA react to cellular response to environmental clues";

"Our findings strongly suggest a causal linkage between the TFs'expression and their targeted lncRNAs' expression." (no, the "targeted lncRNAs" are just putative targets and the link between the expression breadth of TFs and the expression breadth of these lncRNAs could be affected by a selection bias - more details below); "The CLB_TFs preferentially impact the expression of conserved lncRNAs", or "the loss-of-Nrf1 results in larger gene expression changes for conserved lncRNAs than non-conserved lncRNAs, reinforcing its regulatory preference for conserved lncRNA " -> TFs cannot distinguish or "prefer" conserved lncRNAs over non-conserved lncRNAs; a better way to present this is that those lncRNAs that are regulated by some TFs are more evolutionarily conserved.

Response: Thanks for the reviewer's comments. We have revised and updated our statements in our revised manuscript as follows:

- a) *"The conservation of lncRNAs influences the expression of their nearby genes" to "Conserved lncRNAs have a higher expression correlation with their nearby genes".*
- b) *"the influence of lncRNA conservation on their nearby gene expression" to "conserved lncRNAs exhibit a higher correlation only with their nearby gene".*
- c) *"lncRNA conservation may affect how lncRNA react to cellular response to environmental clues" to "To further inspect the functional importance of conserved lncRNA for these diseases".*
- d) *"Our findings strongly suggest a causal linkage between the TFs'expression and their targeted lncRNAs' expression" to "Our findings further suggest that these TFs are likely to regulate the expression of conserved lncRNAs".*
- e) *"The CLB_TFs preferentially impact the expression of conserved lncRNAs" to "The CLB_TFs are more likely to modulate the expression of conserved lncRNAs".*
- f) *"the loss-of-Nrf1 results in larger gene expression changes for conserved lncRNAs than non-conserved lncRNAs, reinforcing its regulatory preference for conserved lncRNA" to "the loss-of-Nrf1 results in larger gene expression changes for conserved lncRNAs than non-conserved lncRNAs, reinforcing that those lncRNAs that are regulated by some TFs are more evolutionarily conserved"*

Regarding the reviewer's concern about the selection bias, we have added more detailed analysis to confirm it below.

- The analyses often lack appropriate controls. For example, the authors confirm that conserved lncRNAs are more broadly expressed than non-conserved lncRNAs. **That likely means that conserved lncRNAs also have on average higher expression levels than non-conserved lncRNAs in the cell types in which they analyze the effect of TF knockdown or overexpression.** Given that the power to detect differential expression between conditions strongly depends on gene expression level, this could easily affect the differences between conserved and non-conserved lncRNAs in terms differential expression after TF knockout or overexpression. It is also more likely that the promoters of conserved lncRNAs are better defined, because full-length gene models are easier to reconstruct for highly expressed genes than for weakly expressed genes. This means that there will be larger frequencies of TF binding sites in these promoters. Overall, gene expression level differences need to be accounted for in all analyses.

Response: Thank you so much for the reviewer's comments. To address the reviewer's concern, we selected a subset of conserved lncRNAs and a subset of non-conserved RNAs with matched abundance to examine the differentially expressed gene between TF knockout/overexpression and control. We found that the percentage of differentially expressed gene are generally higher in conserved lncRNAs than that in non-conserved lncRNAs (Below Figure A, B). Moreover, we observed that density of TF-binding sites is higher in conserved lncRNA than that in non-conserved lncRNA (Below Figure C, D). Therefore, the differential expression and TF-binding sites are not merely due to the expression abundance between conserved and non-conserved lncRNAs.

We also check characteristic including the gene length, transcript number, etc. between the conserved and non-conserved lncRNA with matched abundance across the tissues. The conserved lncRNAs still have longer gene length, more transcripts and exons than the non-conserved lncRNAs, confirming that the effect sizes of our results are not determined by the expression.

We didn't include these figures in our revised manuscripts, because these parts didn't add any new conclusion to our manuscripts.

"Another example is the authors' interpretation of the frequency of disease SNPs associated with neurological disorders - both conserved and non-conserved lncRNAs are associated with such

SNPs, which the authors interpret as "suggesting an involvement of lncRNAs in neuronal development and function.". But what is the overall frequency of SNPs involved in neurological disorders compared to other diseases? It may simply be the case that these diseases have been better studied so far than other diseases."

We agree with the reviewer that the total of disease-associated lncRNAs will bias the frequency of disease-associated lncRNAs in the conserved and non-conserved lncRNAs for each disease. We have reanalysed this part and normalized proportion by dividing the number of disease-associated conserved/non-conserved lncRNA by the total of disease-associated lncRNA and total of conserved/non-conserved lncRNAs. The results show that the lncRNAs involved in neurological disorders are not overall frequency compared to other diseases (Figure 1F). We have updated the statement for this part in the revised manuscript.

Figure legend: Analysis for conserved and non-conserved lncRNAs with matched abundance. (A, B) Percentage of differentially expressed genes between TF overexpression (A) / knockout (B) and control in the conserved and non-conserved lncRNAs with matched abundance. (C, D) Density of TF-binding sites in the promoter of conserved and non-conserved lncRNAs with matched abundance. Transcript number (E), exon number (F), and gene length (G) between the conserved and non-conserved lncRNAs with matched abundance across the tested tissues.

- The text needs to be carefully proofread or even rewritten. There are many grammatical and typographical errors, with some phrases impossible to understand.

Response: Following the reviewer's suggestions and comments, we have rewritten some texts and checked the manuscripts carefully to correct the grammatical and typographical errors. Please see the revised manuscripts.

March 14, 2023

RE: Life Science Alliance Manuscript #LSA-2023-02002-TR

Dr. Lei Sun
Duke NUS Graduate Medical School
CVMD
8 College Road
Singapore 169857

Dear Dr. Sun,

Thank you for submitting your revised manuscript entitled "Multi-dimensional conservation analysis decodes the expression of conserved long noncoding RNAs". We would be happy to publish your paper in Life Science Alliance pending final revisions necessary to meet our formatting guidelines.

- please upload your main manuscript text as an editable doc file
- please add ORCID ID for both corresponding authors-you should have received instructions on how to do so
- please add the Twitter handle of your host institute/organization as well as your own or/and one of the authors in our system
- GEO dataset GSE156270 should be made publicly accessible at this point

A. FINAL FILES:

B. MANUSCRIPT ORGANIZATION AND FORMATTING:

Sincerely,

March 21, 2023

RE: Life Science Alliance Manuscript #LSA-2023-02002-TRR

Dr. Lei Sun
Duke NUS Graduate Medical School
CVMD
8 College Road
Singapore 169857

Dear Dr. Sun,

Thank you for submitting your Research Article entitled "Multi-dimensional conservation analysis decodes the expression of conserved long noncoding RNAs". It is a pleasure to let you know that your manuscript is now accepted for publication in Life Science Alliance. Congratulations on this interesting work.

DISTRIBUTION OF MATERIALS:

Again, congratulations on a very nice paper. I hope you found the review process to be constructive and are pleased with how the manuscript was handled editorially. We look forward to future exciting submissions from your lab.

Sincerely,
